# Oral Administration of a Novel, Synthetic Ketogenic Compound Elevates Blood β-Hydroxybutyrate Levels in Mice in Both Fasted and Fed Conditions

**DOI:** 10.3390/nu16203526

**Published:** 2024-10-18

**Authors:** Maricel A. Soliven, Christopher Q. Rogers, Michael S. Williams, Natalya N. Thomas, Edward Turos, Dominic P. D’Agostino

**Affiliations:** 1Laboratory of Metabolic Medicine, Department of Molecular Pharmacology and Physiology, Morsani College of Medicine, University of South Florida, Tampa, FL 33612, USA; solivenma.inc@gmail.com (M.A.S.); natalya071001@gmail.com (N.N.T.); ddagosti@usf.edu (D.P.D.); 2Department of Chemistry, University of South Florida, Tampa, FL 33620, USA; msw10g@gmail.com (M.S.W.); eturos@usf.edu (E.T.)

**Keywords:** ketogenic diet, ketosis, exogenous ketones, hyperketonemia, hypoglycemia, ketone metabolic therapy, nutritional supplements, medium-chain triglycerides, diabetes

## Abstract

Background/Objectives: Elevating ketone levels with therapeutic nutritional ketosis can help to metabolically manage disease processes associated with epilepsy, diabetes, obesity, cancer, and neurodegenerative disease. Nutritional ketosis can be achieved with various dieting strategies such as the classical ketogenic diet, the modified Atkins diet, caloric restriction, periodic fasting, or the consumption of exogenous ketogenic supplements such as medium-chain triglycerides (MCTs). However, these various strategies can be unpleasant and difficult to follow, so that achieving and sustaining nutritional ketosis can be a major challenge. Thus, investigators continue to explore the science and applications of exogenous ketone supplementation as a means to further augment the therapeutic efficacy of this metabolic therapy. Methods: Here, we describe a structurally new synthetic triglyceride, glycerol tri-acetoacetate (Gly-3AcAc), that we prepared from glycerol and an acetoacetate precursor that produces hyperketonemia in the therapeutic range (2–3 mM) when administered to mice under both fasting and non-fasting conditions. Animal studies were undertaken to evaluate the potential effects of eliciting a ketogenic response systemically. Acute effects (24 h or less) were determined in male VM/Dk mice in both fasted and unfasted dietary states. Results: Concentration levels of β-hydroxybutyrate in blood were elevated (βHB; 2–3 mM) under both conditions. Levels of glucose were reduced only in the fasted state. No detrimental side effects were observed. Conclusions: Pending further study, this novel compound could potentially add to the repertoire of methods for inducing therapeutic nutritional ketosis.

## 1. Introduction

Elevated levels of ketone bodies in the blood (ketonemia) or urine (ketonuria) occur as a consequence of a specific metabolic state known as ketosis. Pathological ketosis occurs in various conditions including diabetes and acute alcohol intoxication and typically presents as a medical emergency. Conversely, nutritional ketosis is positively associated with numerous health benefits and can be achieved by various means including the classical ketogenic diet, the modified Atkins diet, caloric restriction, periodic fasting, or consumption of exogenous ketogenic supplements such as medium-chain triglycerides (MCTs) [1,2,3,4,5]. It is historically significant that nutritional ketosis has been utilized for more than 100 years as an effective treatment for drug-resistant seizures [6] and has been shown to be effective in numerous models of seizure [7,8,9,10]. More recently, interest in the ketogenic diet has been prompted by its potential effectiveness as a treatment for obesity [11], cancer [12], and neurological diseases such as Alzheimer’s disease [13]. Development and use of ketone metabolic therapy occurred over the past decade and represent an attractive alternative to circumvent the restrictive dietary approach conventionally used to induce and sustain therapeutic ketosis.

Human diets typically provide fuel for producing energy from three macronutrients: fats, proteins, and carbohydrates. Energy derived from cleaving the carbon bonds in these compounds is used to generate ATP and the cellular reducing agents NADPH and NADH. Fats are the most energy-dense of these compounds per unit mass at 9 kCal/g, while proteins and carbohydrates each contain 4 kCal/g. Dietary fats are primarily derived from triglycerides, which are structures comprised of a glycerol molecule bound by ester bonds to three fatty acids comprised of long aliphatic chains typically having lengths between 14 and 22 carbons. Monoglycerides and diglycerides are also naturally occurring fat sources but are less abundant than triglycerides.

The ketone bodies β-hydroxybutyrate (βHB), acetoacetate (AcAc), and acetone are considered a potential fourth orally consumable macronutrient that represents an alternative energy source, with high uptake in the heart and the brain. Ketones are generated endogenously via hepatic ketogenesis, which results in the condensation of the acetyl CoA molecules that are produced by β-oxidation of fatty acids. The R-enantiomer of β-hydroxybutyrate (R-BHB) is used as a proxy measure of blood ketones. This is commonly performed by individuals who wish to monitor their blood ketones by use of blood test strips and commercially available, hand-held meters, similar to point-of-care glucose meters used by individuals with diabetes. Measured this way, baseline blood ketones in humans consuming a typical western diet are usually 0.1 to 0.2 mM. Therapeutic ketosis can be reached at blood levels of 0.5 to 2.5 mM, which can be achieved by a variety of means including fasting, calorie restriction, carbohydrate-restricted ketogenic diets, medium-chain triglycerides (MCTs), and consuming exogenous ketone supplements [14]. This level of ketosis is in contrast to pathological ketoacidosis subsequent to diabetes or ethanol intoxication where R-BHB levels of 5 to 12 mM are elicited, fasting glucose levels can exceed 300 mg/dL, and blood pH levels can approach 7.0 [15], typically in the context of hyperglycemia. Currently, there are numerous products available in the consumer market that are promoted as ketogenic supplements. These include medium-chain triglycerides (MCTs), ketone salts, and ketone esters. MCTs are made up of glycerol molecules connected to fatty acid chains between 8 and 12 carbons long. Ketone salts are predominantly made up of BHB ionically bonded to sodium, calcium, potassium, or magnesium. Supplemental, nutritional ketone esters have consisted of various combinations of molecules such as 1, 3-butanediol, BHB, and AcAc joined via either one or two ester linkages. Specifically, investigators have shown the promise of elevating blood ketones to therapeutic levels with butanediol acetoacetate mono- and di-esters [16] and a BHB-AcAc mono-ester [17]. See Figure 1 for relevant chemical structures.

Here, we describe the effects of an orally delivered, novel, ketogenic compound on the levels of blood glucose and ketones in mice. The compound (IUPAC Name: propane-1,2,3-triyl tris (3-oxobutanoate)) can be described structurally, as consisting of a glycerol backbone bound to three molecules of acetoacetate via ester bonds on each of the three glycerol carbons, yielding a common name of glycerol tri-acetoacetate (Gly-3AcAc).

## 2. Materials and Methods

### 2.1. Ethical Declaration

All animal procedures were performed in compliance with the fNIH Guide for the Care and Use of Laboratory Animals. Animals were used with permission of, and under the guidance of, the USF Institutional Animal Care and Use Committee (IACUC # IS00005524).

### 2.2. Chemistry 

#### 2.2.1. Synthesis of Propane-1, 2, 3-Triyl Tris (3-Oxobutanoate) (**1**)

Glycerol (1 eq.) and t-butyl acetoacetate (3.1 eq.) (both from Sigma-Aldrich, St. Louis, MO, USA) were placed in a round-bottomed flask equipped with a Dean–Stark trap and condenser, and the mixture was heated with stirring to 150 °C. After distillation of t-butanol (3 eq.) was complete, the reaction mixture was removed from heat, and excess t-butyl acetoacetate was removed via rotary evaporation to provide the title compound **1** as an oil. This compound is a water-soluble, clear, light yellow, nearly colorless, viscous liquid with a density of 1.1 g/mL and a molecular mass of 344 Da (Figure 2). ^1^H NMR analysis showed the appearance of signals for the product with only a trace amount of residual t-butyl acetoacetate (Appendix A). Proton NMR protocol: ^1^H NMR (400 MHz, Chloroform-*d* (Sigma-Aldrich, St. Louis, MO, USA)) δ 5.38–5.28 (m, 1H), 4.37 (d, *J* = 4.2 Hz, 1H), 4.34 (d, *J* = 4.2 Hz, 1H), 4.28 (d, *J* = 6.2 Hz, 1H), 4.25 (d, *J* = 6.2 Hz, 1H), 3.43 (s, 6H), 2.17 (s, 9H). NMR analysis was performed using a Bruker Neo 400 MHz spectrometer (Billerica, MA, USA).

#### 2.2.2. Liquid Chromatography–Mass Spectrometry

Samples were processed for the detection and quantification of AcAc at the Metabolic Phenotyping and Mass Spectrometry Core at the University of Tennessee, Health Science Center in Memphis, TN, USA. Whole blood was collected, stabilized with cold 0.2 mol/L NaB2H4 (Sigma-Aldrich, St. Louis, MO, USA) to convert AcAc into [2-2H] BHB, and then immediately frozen on dry ice. Samples were stored at 80 °C until analyses. Briefly, internal standard of [2,4-13C2] BHB was added to the treated blood samples (15 μL), and BHB was extracted and converted to its trimethylsilyl (TMS) derivative by reacting lyophilized sample with 80 μL of bis (trimethylsilyl) trifluoroacetamide + 10% trimethylchlorosilane (Regis, Morton Grove, IL, USA) for 30 min at 75 °C. di-TMS derivative of BHB was analyzed by gas chromatography–mass spectrometry (GC–MS) using a 5973 mass spectrometer, (Agilent Technologies, Santa Clara, CA, USA) linked to a 6890 gas chromatograph equipped with an autosampler. Briefly, BHB (M/Z 233) was detected under the condition of electron ionization (EI) mode. M1 ion (M/Z 234) corresponding to [2-2H] BHB represented the AcAc amount present in the sample after appropriate background subtraction. M2 ion (M/Z 235) corresponded to an internal standard and was used to calculate AcAc and BHB sample concentrations.

### 2.3. Animal Experiments

All VM/Dk mice were obtained from our onsite breeding colony (USF IACUC protocol IS00009383). For all experiments, male VM/Dk mice between the ages of 3 and 5 months were used.

A series of four separate pharmacokinetic experiments were performed. In each experiment, the mice were allocated into groups of 5, such that the body weight of each group was optimally uniform. The size of each group for each experiment consisted of *n* = 5, yielding an E-value of 16 to optimize results with minimal animal resources [18]. Mice were individually housed, provided with igloos and nestlets, and allowed to acclimate to their environment in the vivarium for 1 week. For three of the experiments, food restriction was implemented (i.e., the mice were prevented from access to any food for the indicated time frame). For one of the experiments, mice were given ad libitum access to food. For all four experiments, free access to water was not interrupted. Mice were observed throughout each experiment and daily thereafter to detect any signs of adverse effects. Specifically, observers looked for unnatural mouse behavior such as hunched posture, sluggish movements, shaking, lack of exploratory behaviors, and altered facial expressions. If animals had exhibited signs of animal discomfort and/or impairment, they would have been removed from the study and euthanized. Mice were noticed to all behave normally and were active and mobile. They each accessed and drank their water and consumed food in those cases where food restriction was not implemented.

Each mouse received an oral gavage (with a 25-gauge gavage needle) of the indicated compound. At baseline, and over the course of the experiment, capillary blood samples were obtained via the tail snip method [19]. At selected timepoints, blood glucose and R-BHB were determined by use of Precision Xtra blood glucose and ketone monitoring system (i.e., “test strips”, Precision Xtra, Abbott Laboratories, Abbott Park, IL, USA). See Table 1 for all relevant experimental parameters. 

The concentration for maximum effect (C_max_) of BHB was calculated as the mean of the maximum concentration of BHB measured in each subject over the course of the experiment.

#### 2.3.1. Experiment 1: Dose Response

A 24 h dose–response experiment was performed with periodic measurements of blood glucose and BHB. Each group of mice received oral gavages of Gly-3AcAc in one of the following doses (in mg/g): 0 (this was the negative control group which received a water gavage), 2.5, 5.0, or 7.5. Two hours prior to administration of the gavage, food restriction was implemented. Free access to water was not interrupted. Access to food was reinstituted after the 5 h timepoint.

#### 2.3.2. Experiment 2: High Dose, Fasted for 24 h

A 24 h experiment was performed comparing a negative control group, which received an oral water gavage, and a group receiving the 7.5 mg/g dose with periodic measurements of blood glucose and BHB. Two hours prior to administration of the gavage, food restriction was implemented. Free access to water was not interrupted. Access to food was reinstituted after the 24 h timepoint.

#### 2.3.3. Experiment 3: High Dose, Fasted for 4 h

A 4 h experiment was performed comparing a negative control group, which received an oral gavage of water, and a test group, which received the 7.5 mg/g dose with periodic measurements of blood glucose and BHB. Two hours prior to administration of the gavage, food restriction was implemented. Free access to water was not interrupted. Animals were euthanized at 4 h post-gavage by CO_2_ inhalation and exsanguinated via cardiac puncture. Blood was drawn via cardiac puncture. Serum was separated by centrifugation and stored at −70 °C and was preserved for analysis by mass spectrometry to determine levels of BHB and AcAc. Serum was subsequently analyzed by Metabolic Phenotyping and Mass Spectrometry Facility (Memphis, TN, USA) with an Agilent-7000C Triple Quadrupole GC/MS.

#### 2.3.4. Experiment 4: High Dose, Not Fasted

A 24 h experiment was performed comparing a negative control group, which received an oral gavage of water, and a test group receiving the 7.5 mg/g dose with periodic measurements of blood glucose and BHB. For this experiment, food restriction was not implemented.

### 2.4. Statistical Analysis and Graphical Presentation

For pharmacokinetic analysis, values for serum glucose and BHB at each timepoint were averaged and plotted. The inclusion requirement was that animals exhibit normal behavior throughout experiments. The exclusion criteria for data were that data points greater than 2.5 standard deviation from the mean would be excluded. There were no outliers. No animals or data points were excluded from the analysis. Researchers were not blinded to group allocation. Confounders were not controlled. All data are presented as the mean, with standard error of the mean (SEM). Dose–response data were analyzed using two-way ANOVA repeated measures with Dunnett’s multiple comparisons post hoc test. Data were analyzed, and figures were created using Graph Pad Prism v 6.11. Results were considered significant when *p* < 0.05. For all line graphs representing data, horizontal bars or asterisks indicate timepoints when treatment and control measurements are significantly different from each other (*p* < 0.05). For all column graphs, groups of data points with shared alphabetic symbols are not significantly different from each other.

## 3. Results

### 3.1. Experiment 1: Dose Response, Fasted for 5 h Post-Gavage

Referring to Figure 3, panel A shows that oral gavage of Gly-3AcAc decreased blood glucose in a dose-dependent manner. Administration of water gavage to the control animals resulted in a significant elevation of blood glucose noted at 1 h post gavage, likely due to a stress response [20]. Apparently, the Gly-3AcAc attenuated the glucose rise observed due to the stress effect of the gavage. Consequently, all doses of Gly-3AcAc resulted in a decrease in glucose compared to control at the 1 h timepoint.

Treatment with 2.5 g/kg dose did not significantly alter blood glucose compared to baseline. Treatment with the middle dose decreased blood glucose compared to baseline from 2 through 4 h. Treatment with the high dose decreased blood glucose compared to baseline from 2 through 5 h. The blood glucose of all animals returned to baseline by 24 h post-gavage (Figure 3).

As shown in panel B, oral gavage of Gly-3AcAc increased blood BHB in a dose-dependent manner. Administration of water gavage to the control animals had no apparent effect on blood levels of BHB. Treatment with the low dose of Gly-3AcAc resulted in elevated blood BHB compared to baseline at the 1 h timepoint. Treatment with the middle dose of Gly-3AcAc elevated blood BHB compared to baseline from 2 through 4 h. Treatment with the high dose (7.5 g/kg) elevated blood BHB from 2 to 5 h. The blood BHB of all animals returned to baseline by 24 h post-gavage.

Panel C shows the blood glucose ketone index (GKI), which is a useful tool for assessing ketogenic effects. The GKI is defined as the ratio of glucose (mM) to BHB (mM), and it has been suggested that levels at or below 1 are indicative of beneficial, anti-cancer, therapeutic levels [21]. The GKIs of each treatment group were significantly less than the control group as early as 30 min and up until 5 h post-gavage. The lowest GKIs (less than 1) were seen with the 7.5 g/kg dose at 3 and 4 h post-gavage. The GKI of all animals returned to baseline by 24 h post-gavage.

Panel D shows the BHB C_max_ which is a measure of the maximum concentration of BHB achieved for each individual mouse, independent of the timepoint. There were differences in the kinetics of the individual treatment subjects. For some mice, this occurred as early as 30 min, and for others as late as 5 h. For most mice, it was seen between 3 and 4 h. The BHB C_max_ significantly increased above the control group in a dose-dependent manner. See Appendix A showing C_max_ values and times for each individual subject for experiments 1, 2, and 4.

### 3.2. Experiment 2: High Dose, Fasted for 24 h Post-Gavage

Referring to Figure 4, as shown in panel A treatment with the high dose of Gly-3AcAc resulted in a decrease in blood glucose for up to 8 h which then normalized at 12 h and for the remainder of the experiment. As seen in panel B, this treatment resulted in an elevation of blood BHB for up to 8 h post-gavage which also normalized at 12 h. However, by 16 h, the control group shows an elevation of BHB over the treatment group. Panel C shows the GKI, which for the Gly-AcAc group was significantly lower than the water gavage group from the first post-baseline measurement, until 8 h. However, from 20 h and for the remainder of the experiment, the control group had a significantly lower GKI than the treatment group. While the treated group had higher BHB levels early on (from half an hour until 8 h post-gavage), the control group had higher BHB levels later in the experiment (from 16 h until 24 h post-gavage) at levels similar to that observed by others under similar conditions [22]. As shown in panel D, there was no significant difference in the BHB C_max_.

### 3.3. Experiment 3: High Dose, Fasted for 4 h Post-Gavage

In experiment 3, as shown in Figure 5, panel A, treatment with a high dose of Gly3-AcAc resulted in a decrease in blood glucose for the entire experiment. Likewise, as seen in panel B, blood BHB was elevated for the entire experiment. Using samples obtained at the 4 h time point, LCMS was performed to measure the amount of BHB and AcAc, each of which was elevated in the treatment group over the control group. Panel C shows that the BHB in the untreated group was 0.30 (0.07) mM and in the treated group it was 1.85 (0.87) mM. Panel D shows that the AcAc in the untreated group was 0.01 (0.002) mM and in the treated group it was 0.35 (0.11) mM.

### 3.4. Experiment 4: High Dose: Not Fasted

Figure 6 panel A shows that at the first timepoint post-gavage, the blood glucose of the control group dipped slightly, while that of the treatment group increased. This resulted in a significant difference at this timepoint. However, for the remainder of the experiment, there was no observable difference between the control group and the treated group in terms of glucose response. In panel B, however, note that there was a significant elevation of BHB in the treated group starting at the first measurement post-baseline and continuing until 8 h. In panel C, the net effect of these changes is shown in the GKI, which is significantly different between groups at the 0.5 h mark and from 2 h until 12 h post-gavage. In panel D, the BHB C_max_ values are shown, with the treatment group having a significantly higher value than the control.

## 4. Discussion

The purpose of this study was to determine the effects of oral administration of a novel compound on the blood levels of glucose and BHB in mice. While baseline levels of BHB in humans are typically around 0.1–0.2 mM, in the VM/dk mouse model, they range from 0.5 to 1.0 mM [23]. In humans and in mice, elevated ketone levels have been shown to lead to improvement in diseased contexts including epilepsy, diabetes, obesity, cancer, cardiovascular disease, migraine, and neurodegenerative diseases like Alzheimer’s disease and Parkinson’s disease [24]. Indeed, the ketogenic diet is the standard of care for epilepsy and other seizure disorders that are refractory to pharmacotherapy [25]. Additionally, emerging evidence-based applications for the ketogenic diet include psychiatric disorders such as bipolar and anxiety, traumatic brain injury, long COVID-19, and more [26,27,28].

There are various ways to increase ketone levels including the use of extended fasting, intermittent fasting, time-restricted eating, ketogenic diet, and consumption of exogenous, ketogenic supplements. However, each of these strategies elevates ketone levels by overlapping biochemical mechanisms. At a fundamental level, the body turns fats or fat-like molecules, such as AcAc into hydrolyzed molecules that yield the ketone bodies.

The limitations of this study include the use of only a single mouse model. It would be informative to test this compound in a variety of animal models, including different strains and models for the various conditions that therapeutic ketosis can treat, such as epilepsy, diabetes, obesity, cancer, and neurodegenerative disease. Another limitation is that only simple blood data were collected. Since, in the present study, outcome measures were restricted to blood glucose, BHB, and AcAc; future studies should also expand their reach to include other metabolic markers such as insulin, cortisol, and lipid profiles, as well as potential weight changes and behavioral effects in longer term studies.

This synthetic Gly-3AcAc is anticipated to metabolize by initial hydrolytic cleavage of the three ester moieties through the action of gastric esterases, releasing one equivalent of glycerol and three equivalents of AcAc, which are delivered to the liver via the hepatic portal circulation. Glycerol is transformed in a stepwise process to glucose and released into circulation. Some of the AcAc are transformed to BHB and both are released into circulation. Metabolites are delivered to targets such as muscle, brain, and adipose tissue. 

From prior studies, we see that when one consumes an exogenous ketone supplement—whether monoester, diester, or ketone salt—changes in circulating metabolic markers (ketones and glucose) occur [29,30,31]. The blood levels of ketone bodies are increased and the blood levels of glucose are decreased [32]. In the present study using Gly-3AcAc as the ketogenic agent, we have observed that in both fasted and fed conditions, the concentrations of BHB in blood were increased. However, glucose levels were seen to fall only in the fasted states. Based on the findings in this study, it is tempting to speculate that this compound could have beneficial effects on the numerous clinical conditions alluded to above. We could expect that appropriate use of this compound under medical supervision could help control glucose and BHB levels, which could be particularly important in the treatment of diabetes, epilepsy, and obesity.

As previously mentioned, the blood glucose ketone index (GKI) is a useful tool for assessing ketogenic effects. The GKI is defined as the ratio of glucose (mM) to BHB (mM), and it has been suggested that levels at or below 1 are indicative of beneficial, anti-cancer, and therapeutic levels [20]. In each case, whether fasted or unfasted, the GKIs of the treatment group were significantly less than the control group as early as 30 min and up until 5 h post-gavage. This was a dose-dependent effect as shown in Figure 3. Thus, in the present study, therapeutic GKIs below 1 were seen only with the 7.5 g/kg dose at 3 and 4 h post-gavage. It is reasonable to expect that alternative treatment protocols—such as higher and/or more frequent dosages—would increase the duration of therapeutic ketosis.

There were three particularly interesting findings in this study. First, in both the 4 h and 24 h fasted experiments, some of the mice had blood glucose levels drop below 25 mg/dL for up to 3 h, yet there were no signs of distress such as lethargy or shaking. To our knowledge, this has not been previously reported for a supplemental ketogenic compound. Second, in the 24 h fasted experiment, both the treated group and the control group entered ketosis. While the treated group was forced into nutritional ketosis due to the Gly-AcAc, the control group essentially experienced starvation ketosis due to the prolonged absence of food. Thus, the difference in the effects of Gly-3AcAc versus control on BHB C max was evident not in magnitude, but rather in the timing. Third, in the fed experiment, the blood glucose did not drop at all. It is tempting to speculate that the presence of glycerol was involved in this effect. 

While Gly-3AcAc clearly affects blood ketone levels in this mouse model, numerous questions arise that suggest further work is needed. For instance: What are the biochemical and molecular mechanisms by which this compound elevates BHB levels? How does this compound compare to other ketogenic agents? It would be interesting to conduct future experiments with known positive controls for elevating BHB levels, such as MCT oil, ketone salts, or ketone esters. What are the pharmacological and toxicological profiles? Comprehensive pharmacokinetic studies were conducted to elucidate this compound’s absorption, distribution, metabolism, and excretion. Toxicity should be investigated acutely with studies using higher doses of the compound. Also, long-term safety studies will be required to determine if Gly-3AcAc would potentially qualify for generally recognized as safe (GRAS) approval, as has been performed for other ketogenic molecules or formulations [33].

## 5. Conclusions

Ketone metabolic therapy is being explored for a wide range of therapeutic applications, including metabolic and neurological disorders. Therapeutic ketosis, distinctly different from DKA, can be achieved with a variety of methods, and mildly induced hyperketonemia is often associated with lower or stable glycemia. The use of exogenous ketone supplements to optimize glucose and BHB levels expands the list of tools that clinicians have for treating disorders known to be responsive to therapeutic ketosis. Various ketogenic substances that are already in use have been shown to decrease blood glucose while increasing BHB. It appears that the effects of Gly-3AcAc are unique in that this ketogenic supplement seems to only reduce blood sugar while in the fasted state. In the fed state, the blood glucose is apparently unaffected. It is conceivable that judicious application of supplemental ketosis with this compound and in conjunction with other treatments may help in certain clinical settings. Further study is clearly warranted to define the efficacy, safety, and application of this novel ketogenic agent. 

## Figures and Tables

**Figure 1 nutrients-16-03526-f001:**
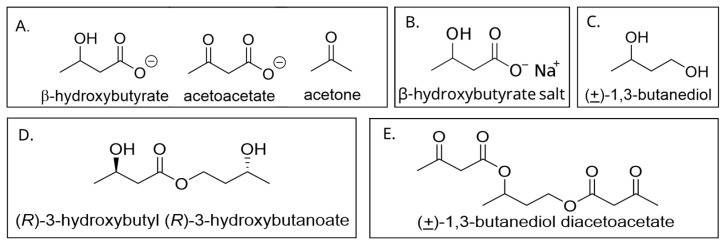
Examples of ketone bodies and relevant exogenous ketones: (**A**) the ketone bodies, β-hydroxybutyrate, acetoacetate, and acetone; (**B**) β-hydroxybutyrate salt of sodium; (**C**) 1,3-butanediol; (**D**) 3-hydroxy butyl 3-hydroxybutanoate ester; and (**E**) 1,3-butanediol diacetoacetate diester.

**Figure 2 nutrients-16-03526-f002:**
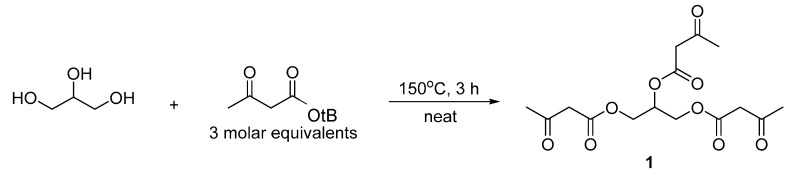
Synthesis of propane-1, 2, 3-triyl tris (3-oxobutanoate) (common name, glycerol tri-acetoacetate (**1**)), abbreviated as Gly-3AcAc.

**Figure 3 nutrients-16-03526-f003:**
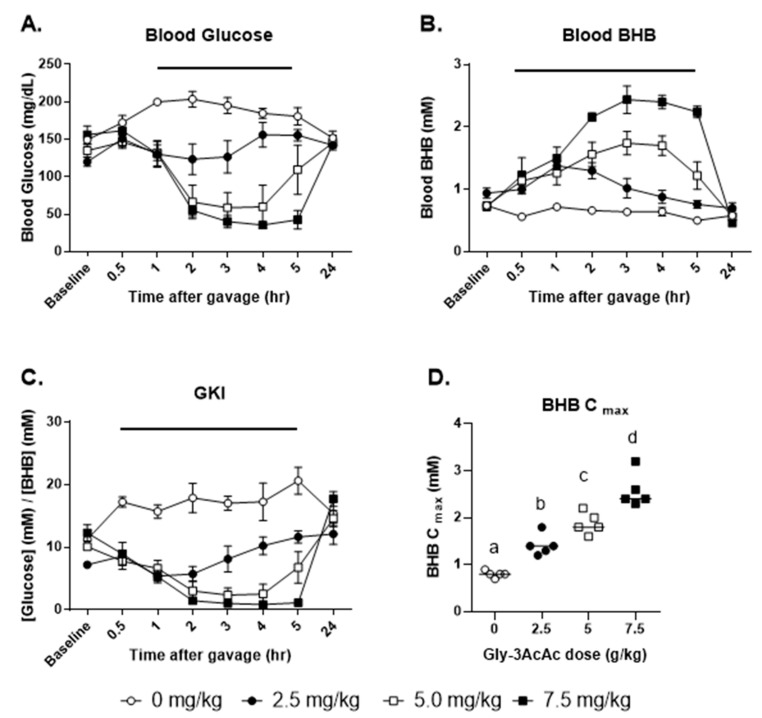
**Experiment 1. Mice were fasted from 2 h before gavage to 5 h after gavage.** (**A**) The average of each group’s blood glucose in mg/dL at each timepoint is shown. For the overall responses, compared to the control group (0 mg/kg), the treatment groups showed significant differences as follows: at 2.5 mg/kg *p* < 0.01; at 5.0 mg/kg *p* < 0.0001; at 7.5 mg/kg *p* < 0.0001. (**B**) The average of each group’s blood BHB in mM at each timepoint is shown. Significant difference from the control group as follows: at 2.5 mg/kg *p* < 0.001; at 5.0 mg/kg *p* < 0.0001; at 7.5 mg/kg *p* < 0.0001. (**C**) The average of each group’s glucose ketone index (GKI) at each timepoint is shown. Significant difference from the control group as follows: at 2.5 mg/kg *p* < 0.0001; at 5.0 mg/kg *p* < 0.0001; at 7.5 mg/kg *p* < 0.0001. (**D**) BHB C_max_, the maximum BHB levels measured for each individual test subject are shown. For all line graphs representing data, horizontal bars indicate timepoints when treatment and control measurements are significantly different from each other (*p* < 0.05). For all column graphs, groups of data points with shared alphabetic symbols are not significantly different from each other.

**Figure 4 nutrients-16-03526-f004:**
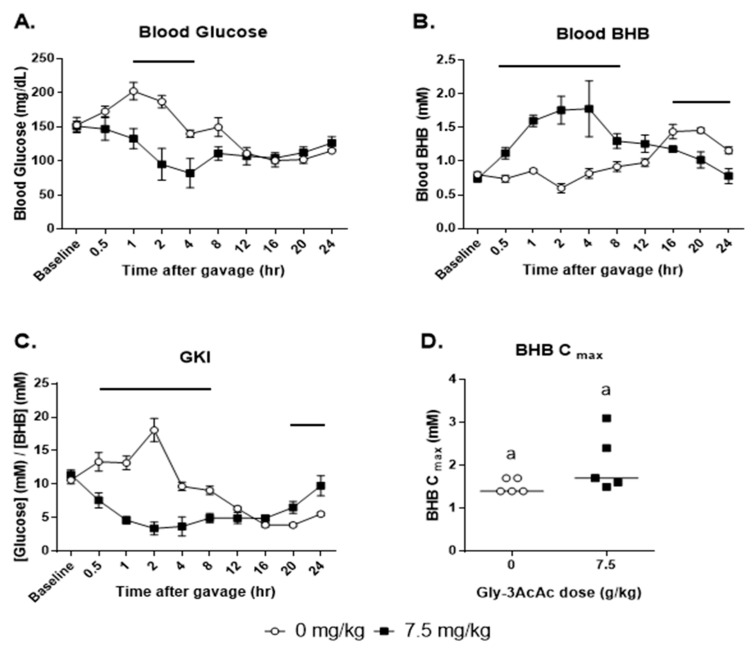
**Experiment 2: Mice were fasted from 2 h before gavage to 24 h after gavage.** Horizontal bars indicate timepoints when treatment and control measurements were significantly different from each other (*p* < 0.05). (**A**) The average of each group’s blood glucose in mg/dL at each timepoint is shown. For the overall responses, compared to the control group, the treatment group was significantly different: *p* < 0.05. (**B**) The average of each group’s blood BHB in mM at each timepoint is shown. For the overall responses, compared to the control group, the treatment group was significantly different: *p* < 0.05. (**C**) The average of each group’s glucose ketone index (GKI) at each timepoint. For the overall responses, compared to the control group, the treatment group was significantly different: *p* < 0.05. (**D**) BHB C_max_, the maximum BHB levels measured for each individual test subject are shown. Groups of data points with shared alphabetic symbols are not significantly different.

**Figure 5 nutrients-16-03526-f005:**
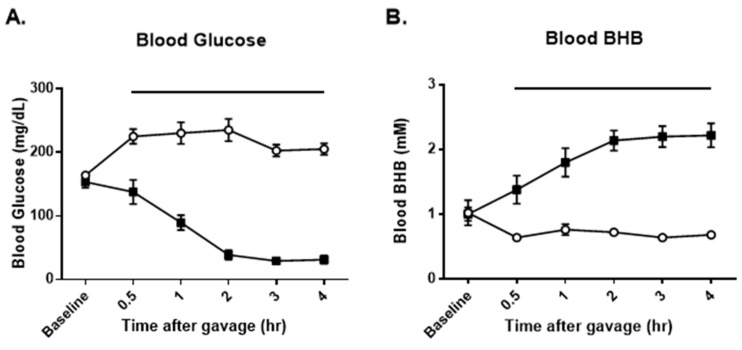
**Experiment 3: Mice were fasted from 2 h before gavage to 4 h after gavage.** (**A**) The average of each group’s blood glucose in mg/dL at each timepoint. (**B**) The average of each group’s blood BHB in mM at each timepoint. (**C**) Serum BHB at 4 h as determined by LCMS. *p* < 0.005. LCMS quantifies both enantiomers of BHB. (**D**) Serum AcAc at 4 h as determined by LCMS. *p* < 0.05. Groups of data points with shared alphabetic symbols are not significantly different.

**Figure 6 nutrients-16-03526-f006:**
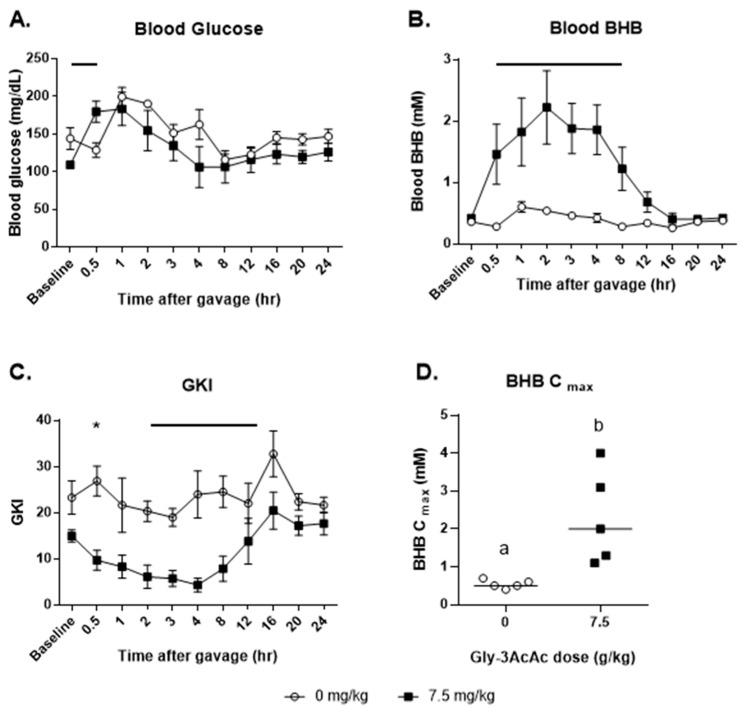
**Experiment 4: Food restriction was not implemented.** (**A**) The average of each group’s blood glucose in mg/dL at each timepoint. (**B**) The average of each group’s blood BHB in mM at each timepoint. (**C**) The average of each group’s glucose ketone index (GKI) at each timepoint. (**D**) BHB C_max_, the maximum BHB levels measured for each individual test subject are shown, *p* = 0.016. Groups of data points with shared alphabetic symbols are not significantly different. For all line graphs representing data, horizontal bars or asterisks indicate timepoints when treatment and control measurements are significantly different from each other (*p* < 0.05).

**Table 1 nutrients-16-03526-t001:** Experimental parameters for each experiment number (#) 1 through 4. (n/a = not applicable).

Exp #	Duration (h)	Doses Used (mg/g)	Food Restriction Relative to Gavage (h)	Outcome Method and Measurements
1	24	2.5, 5, 7.5	2 pre- to 5 post-	Strips, glucose, and BHB
2	24	7.5	2 pre- to 24 post-	Strips, glucose, and BHB
3	4	7.5	2 pre- to 4 post-	LCMS, BHB, and AcAc
4	24	7.5	n/a	Strips, glucose, and BHB

## Data Availability

Original data from Experiments 1–4 is available on the Open Science Framework. https://doi.org/10.17605/OSF.IO/A8SFM.

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
