# Peer review of "Oral Administration of a Novel, Synthetic Ketogenic Compound Elevates Blood β-Hydroxybutyrate Levels in Mice in Both Fasted and Fed Conditions"

_nutrients, 2024, doi:10.3390/nu16203526_

Round 1
Reviewer 1 Report
Comments and Suggestions for Authors
The article detailing the effects of glycerol tri-acetoacetate (Gly-3AcAc) on ketone body levels presents intriguing findings but also has areas that may require further attention and refinement:
1,Long-Term Safety Studies: The study does not provide long-term safety data for Gly-3AcAc. It would be beneficial to know the effects of prolonged use on various organ systems.
2,Pharmacokinetics: Detailed pharmacokinetic data, including absorption, distribution, metabolism, and excretion profiles of Gly-3AcAc, are necessary for a complete assessment of its potential as a therapeutic agent.
3,Physiological effect: Elevating ketone levels with therapeutic nutritional ketosis can help to metabolically manage disease processes associated with epilepsy, diabetes, obesity, cancer, and neurodegenerative disease. A corresponding animal disease model is needed to prove the value and significance of glycerol tri-acetoacetate (Gly-3AcAc).
4,Effector molecule: Whether glycerol tri-acetoacetate (Gly-3AcAc) metabolite β-hydroxybutyrate has a direct effect on the above diseases, it is recommended to design experiments to prove.
5,Broader Metabolic Impact: The study could benefit from a more comprehensive analysis of metabolic parameters, including insulin levels, lipid profiles, and potential weight changes.
6,Clinical Relevance: What are the expected clinical implications of these findings, particularly for conditions like diabetes and epilepsy?
7, Application value: Comparison with Existing Ketone Supplements: How does the efficacy and safety of Gly-3AcAc compare to other exogenous ketone supplements currently available?
8, Mechanistic Insights: The precise biochemical and molecular mechanisms by which Gly-3AcAc elevates BHB levels need further elucidation.
Author Response
Please see attached file with comments and responses.

Reviewer 2 Report
Comments and Suggestions for Authors
In the present research, Soliven et al. tested the effects of a newly synthesized compound, glycerol tri-acetoacetate (Gly-3AcAc), on the blood levels of β-hydroxybutyrate and glucose, in mice. The effects of fasting and different doses of orally administered Gly-3AcAc were assessed. This work could be relevant to the field of therapeutic nutritional ketosis.
The research was correctly performed and the manuscript is, in general, clearly written; however, it needs several revisions, as listed below:
1. In the supplementary materials, please include the data related to the statement "1H NMR analysis showed the appearance of signals for the product with only a trace amount of residual t-butyl acetoacetate." (lines 108-109).
2. In the Materials and methods, please include the 1H NMR and LCMS protocols.
3. In sections 3.1 to 3.4, please mention each panel of Figures 3 to 6 and describe its results separately. Some results from these figures are not described in these sections (for example, Figures 6C-6D are not described in section 3.4).
4. In the discussion, please analyze the main findings of the study in more detail. Please also mention its limitations.
5. In the discussion, when referring to the fasted experiment (lines 289-290), please mention if the fasting was for 4 or 24 hours.
Author Response

(The authors gave the same response as above.)

Reviewer 3 Report
Comments and Suggestions for Authors
Ketogenic diets have potential therapeutic impact on diseases such as epilepsy, diabetes, obesity, cancer and neurodegenerative diseases, but are difficult to implement and maintain. Thus, new approaches to achieve therapeutic ketosis are needed. This is an initial preclinical animal study investigating the effects of an acute dose of a novel ketogenic compound on blood glucose and beta-hydroxybutyrate BHB) in fasted and fed mice. The authors find a dose dependent lowering of blood glucose in fasted mice and dose-dependent rise in BHB in both fasted and fed mice. The manuscript is well-written and data presented clearly. I consider my comments as minor concerns.
1. In the introduction the authors state that baseline ketones measured by blood test strips is usually 0.1-0.2 mM, however, the baseline measurements seem to be around 0.5-1 mM. Can the authors explain this discrepancy? Is it related to species or mouse strain?
2. Figure 3, 4, 5, 6: symbols for significant differences in the graphs A, B and C are needed.
3. Figure 3D: define statistically significant symbols in the figure legend.
4. Figure 3C, 4C, 6C: use a break in the y-axis to visually illustrate that 7.5 mg/kg GKI falls below 1.
5. Figure 4D: traditionally no symbols are used when there is no significant difference.
6. In the abstract line 29-30 the authors state that “No detrimental side effects were observed”, however, no description of the methods used to assess toxicology endpoints are presented nor is any data presented. Please state which potential adverse effects were examined.
7. It would be appropriate in the discussion to mention the interpretation of findings in Figure 4B,D.
8. In the discussion, please discuss the GKI and whether your compound has a therapeutic GKI.
Author Response

(The authors gave the same response as above.)

Round 2
Reviewer 2 Report
Comments and Suggestions for Authors
The authors have properly addressed all my recommendations. Therefore, I support the publication of the revised version of the manuscript.